# Voluntary Activity Modulates Sugar-Induced Elastic Fiber Remodeling in the Alveolar Region of the Mouse Lung

**DOI:** 10.3390/ijms20102438

**Published:** 2019-05-17

**Authors:** Julia Hollenbach, Elena Lopez-Rodriguez, Christian Mühlfeld, Julia Schipke

**Affiliations:** 1Institute of Functional and Applied Anatomy, Hannover Medical School, 30625 Hannover, Germany; Julia.Hollenbach@tiho-hannover.de (J.H.); Lopez-Rodriguez.Elena@mh-hannover.de (E.L.-R.); Muehlfeld.Christian@mh-hannover.de (C.M.); 2Biomedical Research in Endstage and Obstructive Lung Disease Hannover (BREATH), Member of the German Center for Lung Research (DZL), 30625 Hannover, Germany; 3Cluster of Excellence REBIRTH (From Regenerative Biology to Reconstructive Therapy), 30625 Hannover, Germany

**Keywords:** dietary sugar, hyperglycemia, lung mechanics, alveolar septal composition, physical activity, extracellular matrix remodeling

## Abstract

Diabetes and respiratory diseases are frequently comorbid conditions. However, the mechanistic links between hyperglycemia and lung dysfunction are not entirely understood. This study examined the effects of high sucrose intake on lung mechanics and alveolar septal composition and tested voluntary activity as an intervention strategy. C57BL/6N mice were fed a control diet (CD, 7% sucrose) or a high sucrose diet (HSD, 35% sucrose). Some animals had access to running wheels (voluntary active; CD-A, HSD-A). After 30 weeks, lung mechanics were assessed, left lungs were used for stereological analysis and right lungs for protein expression measurement. HSD resulted in hyperglycemia and higher static compliance compared to CD. Lung and septal volumes were increased and the septal ratio of elastic-to-collagen fibers was decreased despite normal alveolar epithelial volumes. Elastic fibers appeared more loosely arranged accompanied by an increase in elastin protein expression. Voluntary activity prevented hyperglycemia in HSD-fed mice. The parenchymal airspace volume, but not the septal volume, was increased. The septal extracellular matrix (ECM) composition together with the protein expression of ECM components was similar to control levels in the HSD-A-group. In conclusion, HSD was associated with elastic fiber remodeling and reduced pulmonary elasticity. Voluntary activity alleviated HSD-induced ECM alterations, possibly by preventing hyperglycemia.

## 1. Introduction

Over the last decades sugar consumption has risen dramatically and in most parts of the world sucrose—a disaccharide composed of glucose and fructose—is the most common sweetener [1]. Glucose is metabolized in every cell of the body and its metabolism is tightly regulated by insulin, whereas fructose is primarily metabolized in the liver promoting de novo lipogenesis [2]. High sucrose and/or high fructose consumption leads to persistent blood hyperglycemia and systemic disorders like metabolic syndrome (MetS) and diabetes type 2 [3].

There is evidence that the lung is a target organ for glucotoxicity-induced complications. MetS defined by hyperglycemia, hypertriglyceridemia, and hypertension among others is a risk factor for diabetes type 2, but also for lung function impairment and lung diseases like chronic obstructive pulmonary disease (COPD) and asthma [4]. In diabetic patients, a poor glycemic control leads to a progressive decline of lung function [5], and respiratory conditions including COPD and obstructive sleep apnea are frequent comorbidities of diabetes [6,7]. Moreover, high blood glucose levels were shown to affect fetal and postnatal lung development by generalized slowing of alveolar septal growth [8]. The underlying mechanistic links between hyperglycemia and lung dysfunction are not entirely understood. A better insight could help to develop novel therapeutic approaches and might identify hyperglycemia as an early risk factor for lung diseases. 

Physical exercise was shown to be effective against a number of diseases including MetS and type 1 and type 2 diabetes, and to reduce blood glucose levels, triglycerides, blood pressure, or waist circumference [9,10]. Moreover, it also exerts beneficial effects on lung function [11]. Generally, exercise has an anti-inflammatory effect by induction of anti-inflammatory mediators such as IL-6, IL-1ra, TNF-R, and IL-10 [12]. In mice which were exposed to cigarette smoke for several weeks, physical exercise reduced oxidative stress in the lung and was therefore able to protect against emphysema development [13]. 

The present study tested the hypotheses, that: (i) prolonged excess dietary sucrose intake affects lung mechanics and structure in mice and that (ii) voluntary activity alleviates these sucrose-induced changes.

## 2. Results

### 2.1. Body Weight and Blood Glucose Concentrations

As already published by Schipke et al. [14], high sucrose diet (HSD)-fed mice had higher body weights (control diet (CD): 40.5 ± 3.6 g, HSD: 45.6 ± 2.0 g, *p* = 0.001) and elevated fasting blood glucose concentrations (CD: 8.4 ± 0.5 mmol/L, HSD: 9.8 ± 1.2 mmol/L, *p* < 0.001) in comparison to CD-fed mice after 30 weeks. Voluntary activity had no impact on the HSD-related body weight increase (CD-active (A): 39.2 ± 4.1 g, HSD-A: 43.4 ± 2.8 g, *p* = 0.045; HSD vs. HSD-A *p* = 0.176), but prevented hyperglycemia in the active HSD-group (CD-A: 8.0 ± 0.5 mmol/L, HSD-A: 8.3 ± 0.3 mmol/L, *p* = 0.373; HSD vs. HSD-A *p* < 0.001).

### 2.2. Lung Mechanics

HSD resulted in lower Elastance H (*p* = 0.015; Figure 1A) and higher static lung compliance (*p* = 0.012; Figure 1B) and inspiratory capacity (*p* = 0.043; Figure 1C) compared to CD. Voluntary activity did not influence the HSD-induced changes in lung mechanics; however, H was reduced in active CD-fed mice compared to non-active CD-fed mice (*p* = 0.03; Figure 1A).

### 2.3. Lung Structure

Left lung volumes were higher in the HSD- as well as in the HSD-A-group compared to their respective controls (CD vs. HSD *p* = 0.039, CD-A vs. HSD-A *p* = 0.002, Figure 2A). This was due to an increase in the parenchyma volume (HSD vs. CD *p* = 0.043, HSD-A vs. CD-A *p* < 0.001, Figure 2B), whereas the non-parenchyma volume was not significantly altered (Figure 2C).

### 2.4. Parenchyma Composition

The parenchymal composition differed between inactive and active HSD-groups (Figure 3A,E). HSD alone induced a higher septal volume (CD vs. HSD *p* = 0.001, Figure 3B) and surface area (CD vs. HSD *p* < 0.001, Figure 3C) compared to controls. In contrast, the combination of activity and HSD resulted in a higher airspace volume (CD-A vs. HSD-A *p* < 0.001, Figure 3F), which was due to increases in both ductal (CD-A vs. HSD-A *p* = 0.001; Figure 3G) and alveolar (CD-A vs. HSD-A *p* = 0.002, Figure 3H) airspace, accompanied by a higher septal surface area (CD-A vs. HSD-A *p* < 0.001, Figure 3C). The thickness of alveolar septa did not differ significantly among the groups (Figure 3D). 

### 2.5. Septal Composition

The increased septal volume in HSD-fed animals was accompanied by volume increases of endothelial cells (CD vs. HSD *p* < 0.001, Figure 4A) and the capillary lumen (CD vs. HSD *p* = 0.003, Figure 4B) compared to CD. Similarly, the absolute volume of interstitial cells (mainly fibroblasts) showed a strong tendency to higher levels (CD vs. HSD *p* = 0.053, Figure 4C). In contrast, the epithelial cell volume was not significantly changed in response to HSD (Figure 4D). Voluntary activity in combination with HSD resulted in elevated absolute volumes of interstitial cells compared to CD-A (*p* = 0.005, Figure 4C). The volume of lipid droplets within interstitial cells was not significantly affected by diet or activity (Figure 4 E).

### 2.6. Extracellular Matrix Composition

Compared to CD, the absolute volumes of the extracellular matrix (ECM) (defined as all non-cellular spaces of the septum which includes proteoglycans, collagen fibers and elastic fibers among others; CD vs. HSD *p* = 0.001, Figure 5A) and of collagen fibers within the ECM (CD vs. HSD *p* = 0.009, Figure 5B) were significantly increased in HSD-fed animals. This was consistent with the higher septal volume in these animals. In contrast, the volume of elastic fibers was unchanged in the HSD-group (Figure 5C) and the ratio of elastic-to-collagen fiber volumes was reduced in HSD compared to CD (*p* = 0.003, Figure 5D). 

The structural appearance of septal elastic fibers differed between animals. They appeared either loosely arranged, densely packed, or showed an intermediate phenotype (Figure 6A). Scoring revealed that elastic fibers of HSD-fed animals resembled the loose phenotype in contrast to the more densely packed elastic fibers in control mice (Figure 6B). Voluntary activity partly changed this categorization. Regarding collagen fibers, there was no morphological difference between the groups.

Next, the parenchymal protein expression of elastin and fibrillin as main components of elastic fibers, and of collagen I and collagen III as most common collagen types within the lung parenchyma (4) was assessed. The elastin expression was markedly increased to about 500% of control levels in response to HSD (CD vs. HSD *p* < 0.001, Figure 6C). Fibrillin protein expression was increased to 200% of control levels (CD vs. HSD *p* < 0.001, Figure 6D), whereas collagen I and III levels were similar to CD (Figure 6E,F). Voluntary activity significantly alleviated the HSD-induced changes in elastin (HSD vs. HSD-A *p* = 0.006, Figure 6C) and fibrillin (HSD vs. HSD-A *p* = 0.014, Figure 6D) expression. 

## 3. Discussion

Excess dietary sucrose intake for 30 weeks resulted in hyperglycemia and lung mechanics alterations indicating reduced elasticity, higher septal volumes, and elastic fiber remodeling. Voluntary activity prevented hyperglycemia and alleviated ECM alterations in HSD-fed mice. Moreover, the parenchymal airspace volume, but not the septal volume, was increased in the HSD-A-group. 

Within the lung, the ECM significantly influences the elastic properties [15]. The ECM is mainly composed of elastic fibers, collagen fibers, and proteoglycans. The molecular components are synthesized by interstitial fibroblasts and released into the interstitium, where they assemble into their fibrillar structure. Collagen and elastic fibers cooperate to create a stable, but also elastic septal architecture [15]. In HSD-fed mice, the collagen fiber volume was higher in accordance with the septal volume increase. Moreover, the protein amount of collagen I and III (as the main collagen types of the lung) within 20 µg total parenchymal protein was similar to control levels. This indicates that the amount and the composition of septal collagen fibers in HSD-fed mice are comparable to control conditions. In contrast, the volume of elastic fibers was similar between HSD- and CD-fed mice despite the higher septal volume in the HSD-group, and the ratio of elastic-to-collagen fiber volumes was reduced. Additionally, the structural appearance of the elastic fibers was different in HSD-fed mice. They appeared loosely arranged with more amorphous material and less electron dense structures in contrast to the more densely packed elastic fibers in the control group. Analysis of the parenchymal protein composition revealed that in response to HSD, the amount of elastin was markedly increased to on average 500% of control levels. Also, the fibrillin amount was higher in HSD-fed mice, although this reached only about 200% of control levels. Thus, the prolonged high dietary sucrose intake influenced the expression of the main elastic fiber components elastin and fibrillin. The altered elastin-to-fibrillin ratio might have caused a divergent composition and thus altered structural appearance of the elastic fibers. Since elastic fibers are mainly responsible for the pulmonary elastic recoil during expiration, this finding could at least contribute to the lower lung elasticity we observed upon HSD feeding. 

Elastic fibers are complex structures and the largest ECM component of the lung. Their extracellular assembly is directed by the fibroblast and requires the coordinated expression of tropoelastin and microfibril components as well as enzymes essential for elastin cross-linking [16]. In other organs like the heart, hyperglycemia-induced effects on fibroblasts are well studied with upregulation of collagen expression as one main effect [17,18,19,20]. In contrast, little is known about glucotoxic effects on pulmonary fibroblasts. Glucose-stimulation of fetal rat lung explants results in greater lipid inclusions within fibroblasts [21], contradicting the unchanged lipid droplet volumes in fibroblasts of HSD-fed mice found in this study. This might be due to differences in age (fetal vs. adult) and/or experimental design (lung explant vs. living organism). In cultured human lung fibroblasts, insulin stimulates collagen synthesis [22], however, direct glucose-related effects on lung fibroblasts are not well studied. Also endothelial cells influence pulmonary ECM composition, either via nitric oxide production or by endothelial-mesenchymal transition, and may therefore contribute to the sucrose-induced changes observed in this study [23].

A functional correlation between hyperglycemia and elastic fiber remodeling could add important insight into the complex and not well understood association between diabetes and pulmonary dysfunction. Abnormal elastic fiber assembly and integrity as a result of genetic mutations is associated with an increased susceptibility for lung diseases [24] and changes in content and composition of ECM components significantly contribute to pathogenesis and progression of asthma, COPD, idiopathic pulmonary fibrosis, pulmonary arterial hypertension, and lung cancer [15,25]. On the other hand, type 1 diabetes mellitus leads to a decrease of lung elasticity [26,27]. Moreover, a poor glycemic control indicated by high HbA1c concentrations in diabetic patients results in a progressive decline in lung function [5] and diabetic individuals are at higher risk of COPD, asthma, pulmonary fibrosis, and pneumonia [28]. While diabetic microangiopathy in the lung is one explanation for these alterations, also structural changes are considered to play a major role [29,30]. In diabetic rats, the relative amounts of collagen, elastin, and basal laminae in the septum are increased [31] and diabetes induction by streptozotocin, in 3 week old rats, results in increased collagen and elastin [32]. Another study examining normally fed and undernourished diabetic rats concludes that experimental diabetes affects lung connective tissue metabolism and breakdown and thereby leads to increases in lung collagen and elastin [33]. In humans, fasting plasma glucose concentrations of 6.1–6.9 mmol/L without or with impaired glucose clearance indicate a prediabetic state, whereas a fasting plasma glucose concentration equal to or higher than 7 mmol/L is defined as diabetes [34]. This is not directly transferable to mice, as plasma glucose concentrations of 5–8 mmol/L are reported for mice under control conditions [14,35]. Severe diabetes upon streptozotocin injection is reflected by blood glucose concentrations greater than 14 mmol/L [35]. The HSD-fed mice examined here had blood glucose concentrations around 10 mmol/L, which was significantly higher than CD-fed mice (8.5 mmol/L), but below the severe diabetes conditions of other studies [35]. Moreover, glucose tolerance measured by an oral glucose tolerance test was unimpaired in the HSD-group [14], pointing to a prediabetic or early diabetic state. Thus, elastic fiber remodeling seems to be an early hyperglycemia-induced pulmonary alteration that might prime the lung for injury or disease. Moreover, the mouse model used in this study is suitable to monitor for early hyperglycemia-induced pulmonary alterations and to test therapeutic options against disease progression. 

One intervention strategy already known to be effective against many metabolic disorders is physical exercise [9]. Here, voluntary activity prevented hyperglycemia in HSD-fed mice [14] in line with others showing improved blood sugar control due to physical exercise in prediabetic or diabetic individuals [36,37]. In active, HSD-fed mice the parenchymal airspace volume, but not the septal volume, was increased and the absolute volume of interstitial cells was higher compared to CD-A. Moreover, the septal amount of elastic fibers, the elastic-to-collagen fiber ratio and the protein expression of elastic fiber components was similar to control levels. This indicates that voluntary activity alleviated the HSD-induced septal remodeling processes, possibly by prevention of hyperglycemia. Although this did not result in normalization of lung mechanics, the increase in airspace volume instead of septal volume in HSD-A may point to a beneficial effect for elasticity. It was shown before that physical activity affects expression and activity of matrix metalloprotease- 2 (MMP-2) and MMP-9 and their inhibitors TIMP-1 and TIMP-2 in human muscle and tendon tissue [38,39] which are involved in degradation and stabilization of ECM components. In line with that, physical exercise has a positive effect on ECM remodeling in patients with diabetes type 2 by influencing expression of MMP-2 and its tissue inhibitor TIMP-2 [40].

## 4. Materials and Methods

### 4.1. Animals and Study Design

All animal experiments were approved by the Local Institutional Animal Care and Research Advisory committee and permitted by the Lower Saxony State Office for Consumer Protection and Food Safety (Reference number 33.14-42502-04-13/1244, approval date 11.09.2013). Male C57BL/6N mice were purchased from Charles River (Sulzfeld, Germany) at the age of five weeks. After one week of acclimatization in the local housing facility, animals were randomly assigned to four groups. Animals were housed separately under 12 h light and 12 h dark cycle and were fed ad libitum either a CD with a carbohydrate:protein:fat ratio of 70:20:10 kcal% containing 7% sucrose (D12450J, Research diets, New Brunswick, NJ, USA) [14] or a HSD with a carbohydrate:protein:fat ratio of 70:20:10 kcal% containing 35% sucrose (D12450B, Research diets) [14]. Some animals had free access to running wheels for voluntary activity resulting in four experimental groups: (1) CD (n = 10), (2) CD-A (n = 9), (3) HSD (n = 9) and (4) HSD-A (n = 7).

For all animals used in this study, basic data (body weight, energy intake, running distance, circulating plasma lipid levels and glucose homeostasis) were recently published in another context and compared with additional experimental groups by 3-Way ANOVA [14]. Therefore, body weights and fasting blood glucose concentrations of the mice in this study were subjected to the statistical analysis used here (2-Way ANOVA and Tukey post-hoc test) and means and p-values are reported in the results section of this paper. 

### 4.2. Lung Mechanics

After 30 weeks, mice were anesthetized with Ketamine (100 mg/kg body weight; Dr. Graeub AG, Bern, Switzerland) and Xylazin (5 mg/kg body weight; Rompun^®^, Bayer, Leverkusen, Germany) via intraperitoneal injection. Afterwards mice were tracheostomized and mechanically ventilated using the Flexivent small animal ventilator (SCIREQ, Montreal, QC, Canada) with a frequency of 150/min and a tidal volume of 10 mL/kg body weight. To prevent spontaneous breathing, 0.8 mg/kg body weight pancuronium bromide (Actavis^®^, Inresa Arztneimittel GmbH, Freiburg, Germany) was injected intraperitoneally. Three different mechanical parameters were assessed: the tissue elastance (H), the static lung compliance (CST) and the inspiratory capacity (IC) as described elsewhere [41,42]. In brief, elastance H was assessed using the broadband forced oscillation technique at a positive end-expiratory pressure (PEEP) of 3 cm H_2_O. Quasi-static pressure volume loops were measured to calculate CST according to the Salazar-Knowls equation and IC was determined by deep inflation of the lung at a pressure of 30 cm H_2_O. 

### 4.3. Lung Fixation, Sampling, and Processing

Right lung lobes were ligated and the left lung was fixed via tracheal instillation at a hydrostatic pressure of 20 cm H_2_O using 1.5% paraformaldehyde (Sigma-Aldrich, St. Louis, MO, USA) and 1.5% glutaraldehyde (Merck, Darmstadt, Germany) in 0.15 M HEPES buffer (Sigma-Aldrich). Right lung lobes were snap frozen and stored at −80 °C. The left lung was kept in fixative for at least 24 h. The left lung volume was determined by fluid displacement (Principle of Archimedes) [43]. Afterwards, systematic uniform random sampling (SURS) was performed [44] and tissue slices were randomly allocated to light microscopy (LM) or transmission electron microscopy (TEM) analysis. 

Slices for LM analysis were embedded in Technovit 7100 (Heraeus Kulzer, Wehrheim, Germany) as described previously [45]. In brief, tissue slices were osmicated for 2 h followed by an overnight incubation in a half saturated aqueous solution of uranyl acetate. After dehydration with ascending acetone concentrations, samples were embedded in Technovit 7100, 1.5 µm thick sections were cut and stained with toluidine blue. 

Slices for TEM analysis were randomly subsampled into 1 mm^3^ tissue blocks and embedded in epoxy resin (Epon^®^, Serva, Heidelberg, Germany) as described previously [45]. In brief, tissue blocks were post-fixed with osmium tetroxide followed by an overnight en bloc incubation with uranyl acetate. After dehydration with ascending acetone concentrations, samples were embedded in epoxy resin and 60 nm ultrathin sections were cut for analysis.

### 4.4. Stereological Analysis—Light Microscopy

The following parameters were determined by design-based stereology at the LM level: Volume densities and total volumes of the parenchyma and non-parenchyma, the alveolar septa and the alveolar and ductal airspace as well as the surface density and the total surface area of the alveolar septa. Three to four tissue slices per animal were digitized using a slide scanner (Axio Scan.Z1, ZEISS, Oberkochen, Germany) and analyzed using the newCast software (Visiopharm, Hørsholm, Denmark). The analyst was blinded for experimental groups throughout the analysis.

Volumes were determined by point counting [44]. For estimating the parenchyma (V(par,lung)) and non-parenchyma volume (V(nonpar,lung)), random fields of view were provided by the Visiopharm software at a sampling fraction of 50% and an objective lens magnification of 5 x. The test grid consisted of 24 points and points hitting the structure of interest (parenchyma and non-parenchyma) as well as points hitting the reference space (lung tissue) were counted. Volume densities (V_V_) were calculated by dividing the sum of the points hitting the structure of interest by the sum of the points hitting the reference space (Equation (1)) and total volumes (V) were calculated by multiplying the volume density by the total volume of the reference space (Equation (2)):V_V_(struct/ref) = ∑P(struct)/∑P(ref)(1)

V(struct,ref) = V_V_ (struct/ref) × V(lung) (2)

For estimating the septal volume (V(sept,par)) and the airspace volume (V(air,par)) within the parenchyma, fields of view obtained with a sampling fraction of 5% and an objective lens magnification of 20 x were analyzed using the same test grid as above. Volumes were calculated as shown in Equations (1) and (2) with septum or airspace representing the structure of interest and the parenchyma representing the reference space. 

For estimating the alveolar (V(alvair,par)) and ductal (V(ductair,par)) airspace volumes within the parenchyma, random images were obtained at a sampling fraction of 5–10% and an objective lens magnification of 10 x using the same test system as above. Volumes were calculated as shown in Equations (1) and (2) with the alveolar or ductal airspace representing the structure of interest and the parenchyma representing the reference space. 

For estimating the septal surface area within the parenchyma (S(sept,par)), random images obtained at a sampling fraction of 5% and an objective lens magnification of 20 x were analyzed. The test grid consisted of two lines and four points with a known length per point (l/p = 34.14 µm). All intersections of the test lines with the septal surface and all points hitting the parenchyma were counted [43] and the surface area was calculated as shown in Equations (3) and (4):S_V_(sept/par) = 2 × ∑I(sept)/(l/p × ∑P(par)),(3)

S(sept,par) = S_V_ (sept/par) × V(par,lung). (4)

The septal thickness (τ(sept)) was calculated as shown in Equation (5) [44].

τ(sept) = 2 × V_V_(sept/par)/S_V_(sept/par)(5)

### 4.5. Stereological Analysis—Transmission Electron Microscopy

The following parameters were estimated by design-based stereology at the TEM level: volume densities and total volumes of epithelial cells, endothelial cells, interstitial cells, the capillary lumen and the extracellular matrix (ECM; defined as all non-cellular spaces of the septum which includes proteoglycans, water, collagen and elastic fibers among others). Moreover, volume densities and total volumes of collagen fibers and elastic fibers within the septal ECM and of lipid droplets within interstitial cells were quantified in a separate analysis. Three tissue blocks per animal were analyzed. At least 90 random images per animal were taken according to SURS standards with a Morgagni 268 microscope (FEI, Eindhoven, Netherlands) at a primary magnification of 14,000 x. Images were analyzed using the STEPanizer stereology online tool [46] by an analyst blinded for experimental groups.

Volume densities were estimated by point counting as described above. For estimation of epithelium, endothelium, capillary lumen, interstitial cells and ECM, the test grid consisted of 16 points and for estimation of collagen fibers, elastic fibers and lipid droplets within interstitial cells the test grid consisted of 400 points. Points hitting epithelial cells (P(epi)), endothelial cells (P(endo)), capillary lumen (P(caplum)), interstitial cells (P(intcell)), ECM (P(ECM)), elastic fibers (P(elast)), collagen fibers (P(coll)) and lipid droplets (P(LD-intcell)) and points hitting the reference space (septum; P(sept)) were counted. Volume densities and total volumes were calculated as described in Equations (1) and (2). 

### 4.6. Scoring of Structural Elastic Fiber Appearance

The structural appearance of elastic fibers was assessed by an analyst who was blinded for experimental groups. At least 90 random images at a primary magnification of 14,000 x per animal were analyzed and elastic fibers were assigned to one of three groups: (i) loosely arranged, more amorphous components, (ii) intermediate, (iii) densely packed, more electron dense, fibrillar structures (Figure 6A). 

### 4.7. Protein Islolation And Western Blot

Lung samples were homogenized with a tissue lyser (Qiagen, Hilden, Germany), proteins were isolated using the NucleoSpin® RNA/Protein Kit (Macherey-Nagel, Düren, Germany) and protein concentration was measured with a protein quantification assay (Macherey-Nagel). 

20 µg proteins per lane were loaded, fractionated by SDS-PAGE (polyacrylamide gel concentrations in Table 1) and transferred to PVDF membranes (Bio-Rad, Hercules, CA, USA). Membranes were blocked (blocking conditions in Table 1), incubated with the primary antibody, washed, incubated with the secondary antibody (antibody details in Table 1), washed and developed using the Clarity Max™ Western ECL Blotting Substrate (Bio-Rad) and the ChemiDocTM Touch Imager (Bio-Rad). Protein bands intensities were assessed with the Image LabTM Software (Bio-Rad), normalized according to the loading control and expressed as percentage of the CD mean of the respective membrane. 

### 4.8. Statistical Analysis

Data were analyzed with Sigma Plot (SYSTAT Software Inc., Erkrath, Germany) by two-way analysis of variance (2-Way ANOVA) followed by Tukey test for pairwise multiple comparisons. *p*-values < 0.05 were considered significant, *p-*values between 0.05 and 0.1 (0.05 < *p* < 0.1) were considered to show a tendency towards significance [47]. Data are expressed as means ± SD or means and individual data points for each mouse. Graphs were created using GraphPad Prism (version 4, GraphPad Software, San Diego, CA, USA) and figures were constructed with Photoshop CS6 (Adobe, San José, CA, USA).

## 5. Conclusions

In conclusion, high sucrose intake induced hyperglycemia and elastic fiber remodeling that resulted in reduced pulmonary elasticity. In contrast, the alveolar epithelium did not show alterations. This might contribute to the hyperglycemia-related decline in lung function observed in patients and could prime the lung for injury and disease. Future studies are needed to shed further light on this correlation between high blood sugar and ECM remodeling. Voluntary activity prevented hyperglycemia and significantly alleviated elastic fiber alterations in HSD-fed mice, indicating that physical exercise is a potent intervention strategy against sugar-induced pulmonary changes.

## Figures and Tables

**Figure 1 ijms-20-02438-f001:**
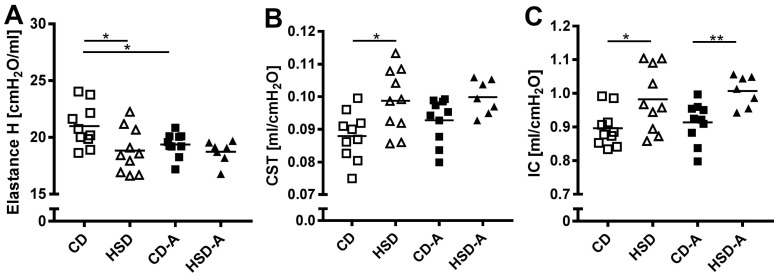
Effects of high sucrose intake and voluntary activity on lung mechanics. Mice were fed a control diet (CD) or a high sucrose diet (HSD) and were left untreated or had access to running wheels (CD-A, HSD-A). Lung mechanics measurements were performed after 30 weeks. (**A**) Elastance H, (**B**) Static lung compliance, (**C**) Inspiratory capacity. Values are individual data points, with means indicated by horizontal lines. Data were compared by 2-Way ANOVA followed by Tukey test; *p*-values < 0.05 are indicated: * *p* < 0.05, ** *p* < 0.01.

**Figure 2 ijms-20-02438-f002:**
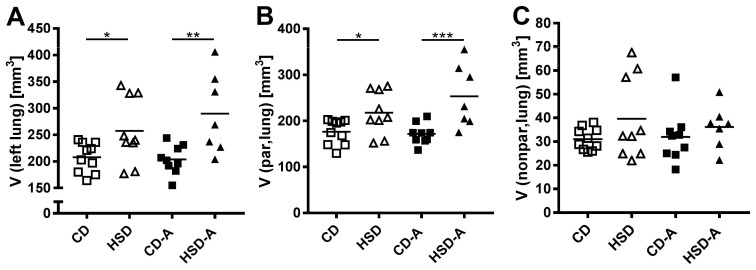
Effects of high sucrose intake and voluntary activity on lung and parenchyma volumes. Mice were fed a control diet (CD) or a high sucrose diet (HSD) and were left untreated or had access to running wheels (CD-A, HSD-A) for 30 weeks. (**A**) Volume of the left lung; (**B**) Volume of left lung parenchyma, (**C**) Volume of left lung non-parenchyma. Values are individual data points, with means indicated by horizontal lines. Data were compared by 2-Way ANOVA followed by Tukey test; *p*-values < 0.05 are indicated: * *p* < 0.05, ** *p* < 0.01, *** *p* < 0.001.

**Figure 3 ijms-20-02438-f003:**
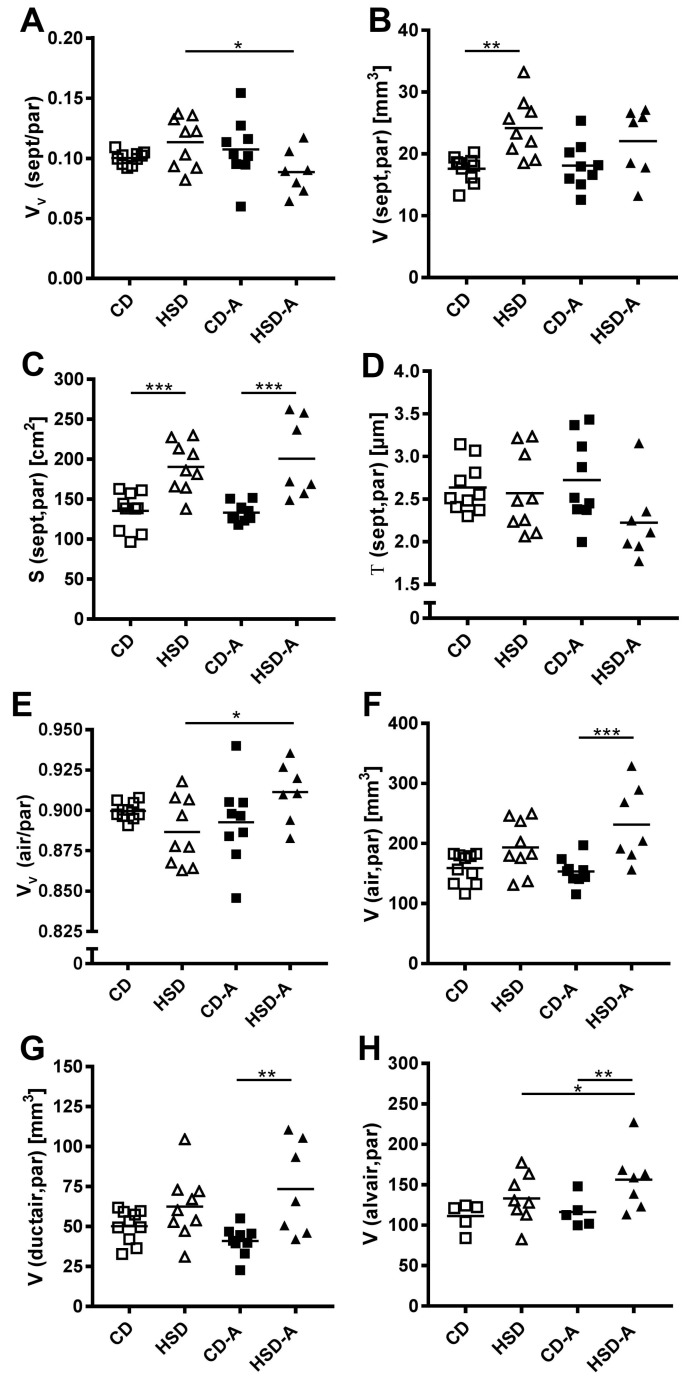
Effects of high sucrose intake and voluntary activity on parenchyma composition. Mice were fed a control diet (CD) or a high sucrose diet (HSD) and were left untreated or had access to running wheels (CD-A, HSD-A) for 30 weeks. (**A**) Septal volume density, (**B**) Septal volume, (**C**) Septal surface area; (**D**) Septal thickness; (**E**) Airspace volume density; (**F**) Airspace volume; (**G**) Ductal airspace volume; (**H**) Alveolar airspace volume. Values are individual data points, with means indicated by horizontal lines. Data were compared by 2-Way ANOVA followed by Tukey test; *p*-values < 0.05 are indicated: * *p* < 0.05, ** *p* < 0.01, *** *p* < 0.001.

**Figure 4 ijms-20-02438-f004:**
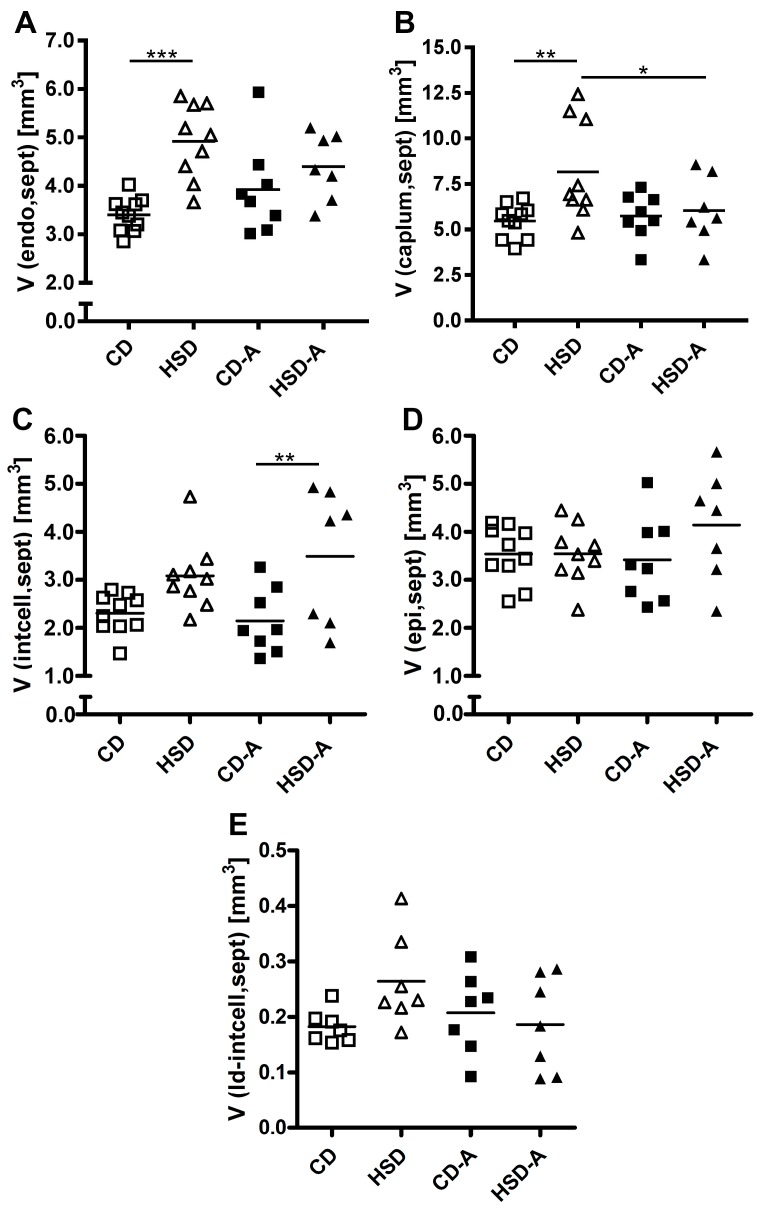
Effects of high sucrose intake and voluntary activity on septal composition. Mice were fed a control diet (CD) or a high sucrose diet (HSD) and were left untreated or had access to running wheels (CD-A, HSD-A) for 30 weeks. (**A**) Volume of septal endothelial cells; (**B**) Volume of septal capillary lumen; (**C**) Volume of septal interstitial cells; (**D**) Volume of septal epithelial cells; (**E**) Volume of lipid droplets within septal interstitial cells. Values are individual data points, with means indicated by horizontal lines. Data were compared by 2-Way ANOVA followed by Tukey test; *p*-values < 0.05 are indicated: * *p* < 0.05, ** *p* < 0.01, *** *p* < 0.001.

**Figure 5 ijms-20-02438-f005:**
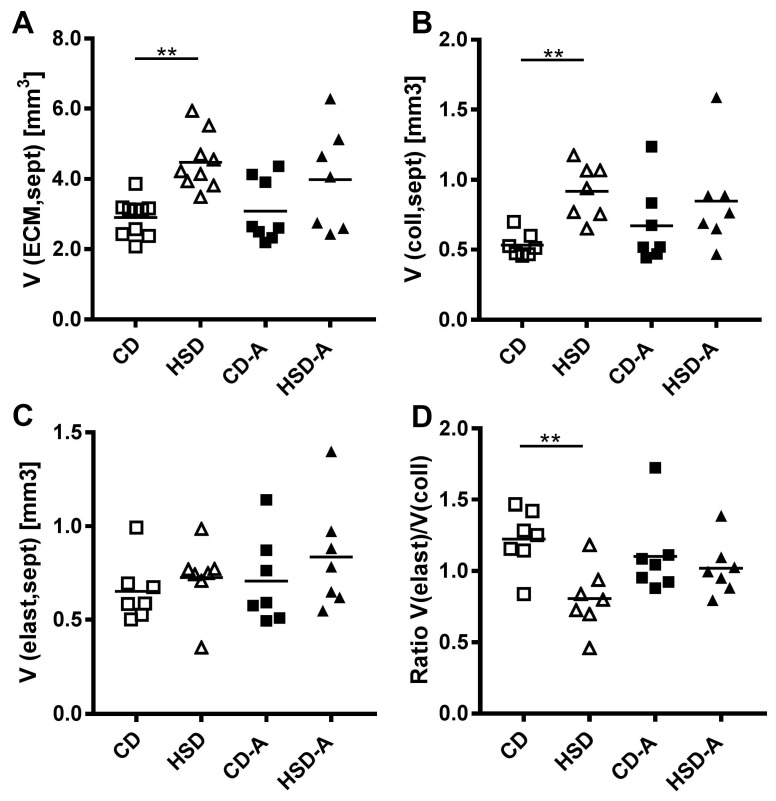
Effects of high sucrose intake and voluntary activity on extracellular matrix composition. Mice were fed a control diet (CD) or a high sucrose diet (HSD) and were left untreated or had access to running wheels (CD-A, HSD-A) for 30 weeks. (**A**) Volume of septal extracellular matrix; (**B**) Volume of septal collagen fibers; (**C**) Volume of septal elastic fibers; (**D**) Ratio of the elastic fiber volume to the collagen fiber volume. Values are individual data points, with means indicated by horizontal lines. Data were compared by 2-Way ANOVA followed by Tukey test; *p*-values < 0.05 are indicated: ** *p* < 0.01.

**Figure 6 ijms-20-02438-f006:**
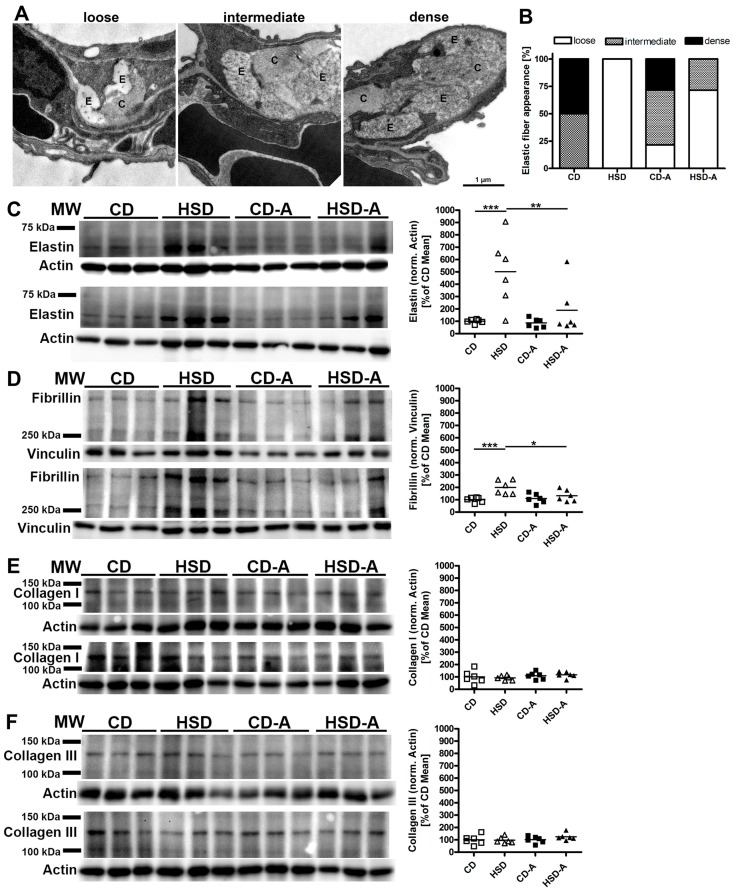
Effects of high sucrose intake and voluntary activity on composition of elastic fibers and collagen fibers. Mice were fed a control diet (CD) or a high sucrose diet (HSD) and were left untreated or had access to running wheels (CD-A, HSD-A) for 30 weeks. (**A**) Representative electron microscopical images showing loose, intermediate or dense structural composition of elastic fibers. E, elastic fibers; C, collagen fibers. (**B**) Scoring of elastic fiber appearance. (**C**) Elastin protein expression; (**D**) Fibrillin-1 protein expression; (**E**) Collagen I protein expression; (**F**) Collagen III protein expression. (**C**–**F**) left: PVDF membranes with protein bands used for quantification, molecular marker bands are indicated; right: protein band intensity signals normalized to the respective loading control and expressed as percentage of the CD mean value of the respective membrane; values are individual data points, with means indicated by horizontal lines. Data were compared by 2-Way ANOVA followed by Tukey test; *p*-values < 0.05 are indicated: * *p* < 0.05, ** *p* < 0.01, *** *p* < 0.001.

**Table 1 ijms-20-02438-t001:** Gel concentrations, blocking conditions, and antibodies used for Western Blot analysis.

Protein	Gel Concentration	Blocking Conditions	Primary Antibody, Dilution, Incubation Conditions	Secondary Antibody, Dilution, Incubation Conditions
**Target Proteins**				
Fibrillin-1	5%	drying and equilibration in 3% BSA for 10 min	Anti-Fibrillin 1 antibody	Peroxidase-AffiniPure F(ab’)2 Fragment Goat Anti-Rabbit IgG
			(ab231094; Abcam, Cambridge, Great Britain)	(111-036-045, Dianova, Hamburg, Germany)
			1:1000	1:20,000
			overnight, 4 °C	1 h, room temperature
Elastin	10%	drying and equilibration in 3% BSA for 10 min	Anti-Elastin antibody	Peroxidase-AffiniPure F(ab’)2 Fragment Goat Anti-Rabbit IgG
			(ab217356, Abcam)	(111-036-045, Dianova)
			1:2000	1:20,000
			overnight, 4 °C	1 h, room temperature
Collagen I	10%	1 h 3% BSA/TBS	COL1A1 (3G3)	m-IgGκ BP-HRP
			(sc-293182, Santa Cruz Biotechnology, Dallas, TX, USA)	(sc-516102, Santa Cruz Biotechnology)
			1:500	1:20,000
			overnight, 4 °C	1 h, room temperature
Collagen III	8%	1 h 3% BSA/TBS	COL3A1 (B-10)	m-IgGκ BP-HRP
			(sc-271249, Santa Cruz Biotechnology)	(sc-516102, Santa Cruz Biotechnology)
			1:500	1:20,000
			overnight, 4 °C	1 h, room temperature
**Loading Controls**				
α-Actin			α-Actin Antibody (G-12)	Peroxidase-AffiniPure F(ab’)2 Fragment Goat Anti-Rabbit IgG
			(sc-130619, Santa Cruz Biotechnology)	(111-036-045, Dianova)
			1:2000	1:20,000
			1 h, room temperature	1 h, room temperature
Vinculin			Vinculin antibody (7F9)	m-IgGκ BP-HRP
			(sc-73614, Santa Cruz Biotechnology)	(sc-516102, Santa Cruz Biotechnology)
			1:2000	1:20,000
			1 h, room temperature	1 h, room temperature

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
