# Peer review of "Voluntary Activity Modulates Sugar-Induced Elastic Fiber Remodeling in the Alveolar Region of the Mouse Lung"

_ijms, 2019, doi:10.3390/ijms20102438_

Round 1
Reviewer 1 Report
This paper discussed the relationship between high glucose diet and lung elastin fibre remodelling in mice. The majority of the data comes from light microscopy, calculating lung mechanics and lung composition. These data concluded that high glucose diet resulted in hyperglycemia and reduced pulmonary elasticity. While voluntary activity prevented hyperglycemia in high glucose diet mice, alleviated elastic fibre remodelling. This paper didn’t study the mechanism of elastic fibre remodelling, but pointed out a possible candidate, fibroblasts. Since endothelial cells can also generate and modulate the extracellular matrix, and they were increased in high glucose diet mice, their involvement in remodelling is also worthy to study.
Author Response
Thank you for your valuable time and useful contribution to our work. We appreciate the input you have given which has definitely helped to improve our manuscript. Please find our point-by-point responses to your comments below.
Point 1: English language and style: Extensive editing of English language and style required.
Response 1: The manuscript has been revised by a native speaker.
Point 2: This paper discussed the relationship between high glucose diet and lung elastin fibre remodelling in mice. The majority of the data comes from light microscopy, calculating lung mechanics and lung composition. These data concluded that high glucose diet resulted in hyperglycemia and reduced pulmonary elasticity. While voluntary activity prevented hyperglycemia in high glucose diet mice, alleviated elastic fibre remodelling. This paper didn’t study the mechanism of elastic fibre remodelling, but pointed out a possible candidate, fibroblasts. Since endothelial cells can also generate and modulate the extracellular matrix, and they were increased in high glucose diet mice, their involvement in remodelling is also worthy to study.
Response 2: We agree with this reviewer that also endothelial cells play an important role in the regulation of pulmonary extracellular matrix composition. Because additional experiments are beyond the scope of this paper, we nevertheless have added the following sentence to the discussion section: “Also endothelial cells influence pulmonary ECM composition, either via nitric oxide production or by endothelial-mesenchymal transition, and may therefore contribute to the sucrose-induced changes observed in this study.”
Reviewer 2 Report
As requested, I reviewed the manuscript (ID ijms-481137) "Voluntary activity modulates sugar-induced elastic fiber remodeling in the alveolar region of the mouse lung", by J. Hollenbach, E. Lopez-Rodriguez, C. Mühlfeld, J. Schipke.
The manuscript deals with the effects of hyperglycemia targeted to lung function. Authors describe the effects of sucrose intake (HSD) on lung mechanics and alveolar septal composition, in mice allowed or prevented to get free access to voluntary activity. Also, molecular analyses of proteins involved in elastic fiber remodeling ECM changings were performed. Overall, authors show that HSD is associated with an impact on elasticity at molecular level as well as pulmonary organ level, and that voluntary activity exerts some counteracting effects on ECM alterations as a consequence of its hyperglycemia preventive ability.
This manuscript supplies an original and effective report on the investigated subject. Methods are comprehensive, with proper references to other research works. Results are clear and abundant (but, in any case, justified), actually, leaving no room for doubt. The Discussion contains all the elements for comprehension of the subject, for understanding the results and for finding hints to deepen information.
The authors demonstrated scientific mastery of their research. My overall comment is actually positive about publishing the research article, with two minor comments:
1) I appreciated the “semi-quantitative analysis of structural elastic fiber appearance” and I think that the representation of the results according to the method is definitely reliable; nevertheless, being absent any reference to possible standardized setup of image acquisition and measurement by the analyst (except magnification), if I were in the authors I would choose a term different than “semi-quantitative” (which usually indicates a normalized analysis with at least one numerical root, even though not absolute).
2) An added value to the results could easily come from ameliorating the size/quality (and framework) of the western blot pictures in figure 6. Visibility is scarce for more than a few bands in lanes; moreover, aiming at disseminating research results, it is always very appreciable to indicate the actual MWs of the bands in your own blots, and possibly of the nearest molecular weight marker (nevertheless, the Table 1 with all Ab details is extremely appreciable).
Thank you for your attention to my opinion.
Author Response
Thank you for your valuable time and useful contribution to our work. We appreciate the input you have given which has definitely helped to improve our manuscript. Please find our point-by-point responses to your comments below.
Point 1: English language and style: English language and style are fine/minor spell check required.
Response 1: The manuscript has been revised by a native speaker.
Point 2: As requested, I reviewed the manuscript (ID ijms-481137) "Voluntary activity modulates sugar-induced elastic fiber remodeling in the alveolar region of the mouse lung", by J. Hollenbach, E. Lopez-Rodriguez, C. Mühlfeld, J. Schipke.
The manuscript deals with the effects of hyperglycemia targeted to lung function. Authors describe the effects of sucrose intake (HSD) on lung mechanics and alveolar septal composition, in mice allowed or prevented to get free access to voluntary activity. Also, molecular analyses of proteins involved in elastic fiber remodeling ECM changings were performed. Overall, authors show that HSD is associated with an impact on elasticity at molecular level as well as pulmonary organ level, and that voluntary activity exerts some counteracting effects on ECM alterations as a consequence of its hyperglycemia preventive ability.
This manuscript supplies an original and effective report on the investigated subject. Methods are comprehensive, with proper references to other research works. Results are clear and abundant (but, in any case, justified), actually, leaving no room for doubt. The Discussion contains all the elements for comprehension of the subject, for understanding the results and for finding hints to deepen information.
The authors demonstrated scientific mastery of their research. My overall comment is actually positive about publishing the research article, with two minor comments:
1) I appreciated the “semi-quantitative analysis of structural elastic fiber appearance” and I think that the representation of the results according to the method is definitely reliable; nevertheless, being absent any reference to possible standardized setup of image acquisition and measurement by the analyst (except magnification), if I were in the authors I would choose a term different than “semi-quantitative” (which usually indicates a normalized analysis with at least one numerical root, even though not absolute).
Response 2: We agree with the reviewer and have renamed the respective analysis “scoring” throughout the manuscript.
Point 3: 2) An added value to the results could easily come from ameliorating the size/quality (and framework) of the western blot pictures in figure 6. Visibility is scarce for more than a few bands in lanes; moreover, aiming at disseminating research results, it is always very appreciable to indicate the actual MWs of the bands in your own blots, and possibly of the nearest molecular weight marker (nevertheless, the Table 1 with all Ab details is extremely appreciable).
Response 3: Thank you for this suggestion. The size of the western blot pictures has been increased and the molecular marker bands have been indicated in figure 6.